Structural insights and characterization of human Npas4 protein

Fahim Ammad 1
Rehman Zaira 1
Bhatti Muhammad Faraz mfbhatti@asab.nust.edu.pk 1
Ali Amjad 1
Virk Nasar 1
Rashid Amir 2
Paracha Rehan Zafar rehanzfr@gmail.com 3
1 Atta-ur-Rahman School of Applied Biosciences (ASAB), National University of Sciences and Technology (NUST) , Islamabad , Pakistan
2 Army Medical College, National University of Medical Sciences , Rawalpindi , Pakistan
3 Research Center for Modeling and Simulation (RCMS), National University of Sciences & Technology (NUST) , Islamabad , Pakistan
Uversky Vladimir
Electronic publication date: 2018 Jun 14
Publication date: 2018
Volume: 6
Electronic Location ID: e4978
Received 2018 Feb 24; Accepted 2018 May 15
Copyright: ©2018 Fahim et al.
Copyright year: 2018
Copyright holder: Fahim et al.
License: This is an open access article distributed under the terms of the Creative Commons Attribution License, which permits unrestricted use, distribution, reproduction and adaptation in any medium and for any purpose provided that it is properly attributed. For attribution, the original author(s), title, publication source (PeerJ) and either DOI or URL of the article must be cited.
License URL: https://creativecommons.org/licenses/by/4.0/

Keywords: Npas4, Transcription factor, Basic loop helix protein (bHLH), Protein-protein interaction, Phylogenetic analysis, Structural prediction

Funding: The authors received no funding for this work.

==============================
Npas4 is an activity dependent transcription factor which is responsible for gearing the expression of target genes involved in neuro-transmission. Despite the importance of Npas4 in many neuronal diseases, the tertiary structure of Npas4 protein along with its physico-chemical properties is limited. In the current study, first we perfomed the phylogenetic analysis of Npas4 and determined the content of hydrophobic, flexible and order-disorder promoting amino acids. The protein binding regions, post-translational modifications and crystallization propensity of Npas4 were predicted through different in-silico methods. The three dimensional model of Npas4 was predicted through LOMET, SPARSKS-X, I-Tasser, RaptorX, MUSTER and Pyhre and the best model was selected on the basis of Ramachandran plot, PROSA, and Qmean scores. The best model was then subjected to further refinement though MODREFINER. Finally the interacting partners of Npas4 were identified through STRING database. The phylogenetic analysis showed the human Npas4 gene to be closely related to other primates such as chimpanzees, monkey, gibbon. The physiochemical properties of Npas4 showed that it is an intrinsically disordered protein with N-terminal ordered region. The post-translational modification analyses indicated absence of acetylation and mannosylation sites. Three potential phosphorylation sites (S108, T130 and T136) were found in PAS A domain whilst a single phosphorylation site (S273) was present in PAS B domain. The predicted tertiary structure of Npas4 showed that bHLH domain and PAS domain possess tertiary structures while the rest of the protein exhibited disorder property. Protein-protein interaction analysis revealed NPas4 interaction with various proteins which are mainly involved in nuclear trafficking of proteins to cytoplasm, activity regulated gene transcription and neurodevelopmental disorders. Moreover the analysis also highlighted the direct relation to proteins involved in promoting neuronal survival, plasticity and cAMP responsive element binding protein proteins. The current study helps in understanding the physicochemical properties and reveals the neuro-modulatory role of Npas4 in crucial pathways involved in neuronal survival and neural signalling hemostasis.

Introduction

NPas4 belong to the basic helix loop helix/per-arnt-sim (BHLH/PAS) transcription factor family (Shamloo et al., 2006). This family is responsible for gearing the expression of target genes on either the positive or negative side (Spiegel et al., 2014). Classically, these transcription factors harbor a DNA binding motif and a dimerisation motif called PAS homology domain (Gu, Hogenesch & Bradfield, 2000). The word ‘PAS’ pertains to the first three proteins of the domain; period (per), aryl hydrocarbon receptor translocator (arnt) and single minded (SIM) (Ooe, Saito & Kaneko, 2009). Functionally, Npas4 is categorized as an immediate early gene (IEG), responsible for direct control of a large number of activity dependent genes that are capable of altering their expression input on the basis of sensory stimuli they receive (Madabhushi et al., 2015; Sun & Lin, 2016). Npas4 is also known for development of glutaminergic and GABAergic synapses in neurons suggesting its crucial role in neuro-circuitary homeostasis and memory formation (Spiegel et al., 2014). Deletion of Npas4 may lead to development of several neuronal plasticity disorders and disorganised management of sensory input (Maya-Vetencourt, 2013). Previous studies consolidated its functional role in cerebral ischemia, memory developmental disorders and related pathologies like autism spectrum disorders, and neuropsychiatric disorders (Choy et al., 2015a; Choy et al., 2015b; Ebert & Greenberg, 2013; Shepard, Heslin & Coutellier, 2017). Limited studies are available which clearly delineate Npas4’s role in regulating its discovered function of neuro-protection and hemostasis in synaptic inter connectivity. The tertiary structure of Npas4 protein along with its physico chemical properties is scarcely available. The current study was mainly aimed to provide the structural insight of Npas4 protein and its possible interaction with other proteins.

Materials and Methods

The flow chart of methodology is shown in Fig. 1.

Figure 1 Flowchart used for functional annotation of Npas4.

Sequence retrieval of Npas4 followed by homology search and multiple sequence alignment and phylogenetic analysis. Assessment of physiological properties, post-translational modifications of Npas4 were predicted using primary sequence. Subsequently, secondary and tertiary structure were predicted through Ab-initio modeling.

Sequence retrieval and homology search

The Npas4 protein sequence was retrieved from UniProt database in FASTA format and was subsequently utilized for homology search using protein BLAST program at NCBI (Blastp). The amino acid sequence of Npas4 was used for homology search using BLASTP at NCBI. One hundred orthologous sequences of Npas4 were retrieved. Out of these 100 sequences 24 full length sequences with homology >97% were selected for phylogenetic analysis.

Multiple sequence alignment and phylogenetic analysis

Multiple sequence alignment of different orthologous sequences were done using ClustalX (version 2.0). All the default paramenters were used for alignment. For Phylogenetic analysis, the tree was constructed by Mega 7 (Kumar, Stecher & Tamura, 2016) using NJ (Neighbour Joining) method along with 1,000 bootstraps. The evolutionary distances were computed using the Poisson correction method. All the other parameters were set as default.

Primary sequence analysis

Protein order-disorder prediction

In order to check whether Npas4 is an intrinsically disordered protein or not, predictions were computed using MobiDB web tool and Meta Disorder Web server (Kozlowski & Bujnicki, 2012; Piovesan et al., 2018). MobiDB predicted the disorder region of proteins through the use of six different tools (DisEMBL, ESpritz, GlobPlot, IUPred, Jronn, VSL2b (Piovesan et al., 2018)). Metadisorder predicts the disorder regions of proteins through the use of 13 different tools and provides the consensus of all these tools (Kozlowski & Bujnicki, 2012). In order to predict the type of disorder in Npas4, charge hydropathy plot and Cumulative Distribution Function analysis were performed through PONDR server (Xue et al., 2010).

Physicochemical analysis of Npas4

The Npas4 primary sequence was used to determine the polarity, mutability, accessibility, bulkiness and refractivity by using Protscale Server on ExPasy platform (http://web.expasy.org/protscale/). The proteins with disorder regions have different composition of flexible and hydrophilic amino acids. The composition of amino acids contribution to flexibility and hydrophilicity was determined through Composition Profiler (http://www.cprofiler.org/).

The signal peptide and cleavage sites were predicted on the basis of artificial neural network through SignaIP-4.1 (http://www.cbs.dtu.dk/services/SignalP/) and SecretomeP (http://www.cbs.dtu.dk/services/SecretomeP/).

Prediction of acetylation and mannosylation sites of Npas4 were tallied through NetAcet web server (http://www.cbs.dtu.dk/services/NetAcet) and NetCGlyc web server (http://www.cbs.dtu.dk/services/NetCGlyc/) respectively, while glycosylation sites were determined through NetOGlyc and NetNGlyc web server (http://www.cbs.dtu.dk/services). Netphos 3.1 server (http://www.cbs.dtu.dk/services/NetPhos) determined the phosphorylation sites for each Thr, Ser and Tyr residues with 0.5 cut off threshold value. NetPhosK 1.0 server (http://www.cbs.dtu.dk/services/NetPhosK) predicted kinase-specific phosphorylation sites in human Npas4. For surface accessibility of predicted phosphorylated sites, NetSurfP program (http://www.cbs.dtu.dk/services/NetSurfP) was utilized. Protparam Server (http://web.expasy.org/protparam/) estimated half life, molecular weight, and amino acid composition of Npas4 protein.

Crystallization propensity of Npas4

Either fully or partially disordered proteins have little tendency to crystallize. The probability of crystallization of Npas4 was reckoned through PPCpred (http://biomine.cs.vcu.edu/servers/PPCpred/), ParCrys (http://www.compbio.dundee.ac.uk/xtal/cgi-bin/input.pl) and CRYSTALP2 (http://biomine.cs.vcu.edu/servers/CRYSTALP2/). ParCrys is based on Parzen Window approach that calculate the probability of crystallization based on hydrophobicity, isoelectric point, and frequencies of Ser, Cys, Gly, Phe, Tyr and Met residues (Overton et al., 2008). PPCpred predicts the probability of protein crystallization on the basis of four steps of crystallization (production of protein material, purification, crystallization, and diffraction-quality crystallization) and for each step it calculates the probability and then by combining all the probabilities it gives the final score of propensity of protein crystallization (Mizianty & Kurgan, 2011).

Prediction of protein binding regions in disorder npas4

It is the property of disordered proteins that they get an ordered conformation upon binding with some globular protein partner. Hence, in an attempt to identify these regions, ANCHOR web server (http://anchor.enzim.hu/) was used. It identifies the regions within the disorder segment that cannot fold and form proper stable structure. ANCHOR utilises the IUPred method for prediction (Meszaros, Simon & Dosztanyi, 2009).

Secondary structure prediction

The secondary structure of Npas4 (Accession No: Q8IUM7) was predicted through GOR4 (Garnier, Gibrat & Robson, 1996) , PsiPred (Jones, 1999) and JPred3 (Cole, Barber & Barton, 2008).

Tertiary structure prediction

Npas4 (Uniprot ID: Q8IUM7) was subjected to BLAST search and less than 25% homology with crytal structure of NPAS3-ARNT complex (5SY7) and NPAS1-ARNT complex (5SY5) was found (Figure S1). Due to the absence of suitable structural homologue, homology modeling cannot be performed. In the absence of structural homologue, threading protein structure prediction approach was used. Threading is a fold recognition method to predict 3D structure of proteins. I-TASSER (http://zhanglab.ccmb.med.umich.edu/I-TASSER) LOMETS (https://zhanglab.ccmb.med.umich.edu/LOMETS/), MUSTER (https://zhanglab.ccmb.med.umich.edu/MUSTER/), SPARSKS-X (http://sparks-lab.org/yueyang/server/SPARKS-X/) RaptorX (http://raptorx.uchicago.edu/) and Phyre 2.0 (http://www.sbg.bio.ic.ac.uk/phyre2/html/page.cgi?id=index) were used for three dimensional structure prediction of human Npas4. LOMETS uses local meta threading approach to predict the tertiary structure of protein. It has nine locally installed threading programs and generates an output on the basis of high scoring target-template alignment. I-TASEER is an iterative threading program that builds 3D structure by using the hierarchical method (Roy, Kucukural & Zhang, 2010). MUSTER is a multisource threading program that works by identifying structural templaes from the protein data bank and then performs the profile-profile alignment and generates 3D structure (Wu & Zhang, 2008). SPARSKS-X is a fold recognition method to predict 3D structure (Yang et al., 2011). Phyre works by searching the structural template for the query sequence followed by multiple sequence alignment and then searching the hidden markov model of structures to find the best structure (Kelley et al., 2015).

Structure validation of Npas4

Validation of three dimensional structure of NPas4 was performed using PROSA (https://prosa.services.came.sbg.ac.at/prosa.php), Qmean (http://swissmodel.expasy.org/qmean/cgi/index.cgi) and Ramachandran plot (http://mordred.bioc.cam.ac.uk/_rapper/rampage.php). Ramachandran plot is the visualization of ψ and φ torsional angles of protein residues and calculates the residues present in favoured, allowed and outlier regions of protein (Lovell et al., 2003). PROSA was used to validate the protein models obtained through different approaches like NMR, X-ray and theoretical calculations (Wiederstein & Sippl, 2007). Qmean provided the estimation of quality of model through Q-score (Benkert, Tosatto & Schomburg, 2008).

Structure refinment

The validated models were then subjected to refinement using Modrefiner server (https://zhanglab.ccmb.med.umich.edu/ModRefiner/). Modrefiner is an atomic, high resolution protein refinement algorithm (Xu & Zhang, 2011).

Protein-protein interaction analysis

String databse (https://string-db.org/) was used to predict the interaction of Npas4 with other cellular peoteins. STRING is a database of known and predicted protein-protein interactions. In order to get the interacting partners of Npas4 the medium confidence interval value of 0.4 was used and 20 maximum interactions for the first shell and 10 maximum interactions for the second shell were used. The interactions may be direct physical interactions or indirect functional associations (Szklarczyk et al., 2017).

Results

Multiple sequence alignment and phylogenetic analysis

Multiple sequence alignment of human Npas4 sequence with other species is shown in Figure S2. The phylogenetic analysis of the aligned sequences of Npas suggests Npas4’s evolutionary conservativeness. The analysis of taxonomic classification of all organisms (Table S1) revealed human Npas4 to be closely related to other primates including chimpanzees (T.N), monkeys and gibbons. The next closest neighbors were rodents (marmots, elephant-shrews, moles, guinea pigs). Other related organisms to rodents were artiodactyla like camels and alpacas. The farthest neighbors of Npas4 sequences were from rodents including rabbits and pikas (Fig. 2).

Figure 2 The phylogenetic tree of human Npas4 generated by NJ method with 1,000 bootstrap value.

The tree showed that human Npas4 closest homology with other primates like chimpanzees (Pan troglodytes), monkeys (Rhinopithecus bieti) and gibbons (Nomascus leucogenys) while the farthest orthologs were rabbits (Oryctolagus cuniculus) and pikas (Ochotona princeps).

Disorder prediction of Npas4

The disorder prediction was carried out using Metadisorder and Mobidb. According to Metadisorder and Mobidb, results indicated Npas4 is an intrinsically disordered protein. According to Metadisorder plot C-terminal half of the protein showed strong disorder while N-terminal of the protein is ordered (position 15–350 a.a). CDF (cumulative distribution function) analysis showed that Npas4 is mixture of ordered and disordered regions as it intersects the boundary (Fig. 3). Primary sequence analysis showed that Npas4 is enriched in disorder promoting amino acids (Pro, Ser, Glu, Gln, Ala and Gly; Table 1). These amino acids have the tendency to prevent the folding of protein (67–70). Pro and Ser have the highest probability. Amino acid sequence analysis of Npas4 protein showed that almost 50% of the protein composed of disorder region. Flexibility analysis of Npas4 was performed using composition profiler and DynaMine server and both servers showed that Npas4 is enriched with flexible amino acids like Pro, Ser and Gln (Fig. 4A). Low hydrophobicity and high hydrophilicity promote disorder in the proteins (Dyson & Wright, 2016). The analysis according to the composition profiler server suggests Npas4 is enriched with hydrophilic amino acids like Gln, Ser, Pro and Thr (Fig. 4B).

Figure 3 Disorder prediction of Npas4.

(A) Metadisorder server predicts the disorder regions in Npas4 primary sequence. The plot shows the prediction of three versions of Metadisorder. The value above 0.5 shows disorder region from position 350 to 802. (B) CDF plot of Npas4 calculated through PONDR server. It predicts either order or disorder proteins. Black line is the boundary, the proteins above the boundary are predicted to be ordered, below the boundary are disordered while the proteins that intersect the boundary is the mixture of order and disorder. According to CDF, Npas4 is mixture of order and disorder.

Physico-chemical properties of Npas4

The physico-chemical properties were determined by Protscale server (Fig. 5). The higher score suggested a higher probability of that particular property of Npas4. The ratio of the side chain volume to the length of an amino acid suggested protein bulkiness and may affect the local structure of a protein. The bulkiness values of Npas4 (Fig. 5A) range from 9.713 (position 731aa) to 18.729 (position 295 aa). The hydrophobicity prediction values by Hopp and wood score (Fig. 5B) were between −0.711 (position 630aa) and 0.422 (position 44 aa). The dipole–dipole intermolecular interactions between the positively and negatively charged particles depicted polarity as predicted through Zimmerman score (Fig. 5C) The predicted score lies between 0.272 (position 81 and 82aa) and 34.216 (position 630 aa). This data showed that Npas4 possesses polarity. Mutability, which is the probability that amino acids will bring a change over a particular evolutionary period of time, was determined through the relative mutability score (Fig. 5D) and the values lie between 46.000 (position 295aa) and 102.889 (position 18 aa) which showed that Npas4 has mutability potential.

Table 1 Disorder promoting amino acids of Npas4.

Disorder promoting amino acids	Order promoting amino acids	
Polar/charged amino acids	Small amino acids	Hydrophobic or bulky	
Pro	10.3%	Ala	7.8%	Leu	13.0%	
Ser	10.2%	Gly	6.6%	Phe	4.9%	
Glu	7.2%			Val	3.2%	
Gln	5.2%			Tyr	2.6%	
Arg	3.4%			Ile	2.5%	
Lys	2.4%			Asn	2.1%	
				Cys	1.6%	
				Trp	0.87%	

Figure 4 Amino acid composition of Npas4.

(A) Flexibility pattern of amino acids in Npas4 calculated through Composition Profiler. Rigid amino acids are shown in green while flexible amino acids are shown in red. (B) Hydrophobicity of amino acids in Npas4 calculated through Composition Profiler. Hydrophilic amino acids are shown in green color while hydrophobic amino acids are shown in black color.

Figure 5 Prediction of physico-chemical properties of Npas4.

X-axis shows amino acid sequence from N- to C-terminal while Y-axis shows scores computed by each algorithm. (A) bulkiness; (B) hydrophobicity; (C) polarity; (D) relative mutability.

The signal peptide of Npas4 was computed using SignalP Server. The presence of signal peptide was measured through C- (raw cleavage site score), S- (signal peptide score) and Y-score (combined cleavage site score). S-score is the estimation of possible signal peptide while D-score is the average of mean S and the max Y-score and its discriminate signal peptide from non-signal peptides. In Npas4, the D-score was 0.450 which was less than the cut off value of 0.5 which showed absence of signal peptide (Fig. 6). The secretory nature of Npas4 was predicted through SecretomeP and the NN score which was 0.5.

Figure 6 Prediction of signal peptide of Npas4 by SignalP server.

X-axis represents amino acid sequence from N- to C- terminal while Y-axis represents scores. Purple line shows the threshold value of 0.5. The proteins having C, S and D scores above threshold show the probability of having the signal peptide.

Crystallization propensity of Npas4

Crystallization propensity was predicted through multiple resources(name) inorder to get a validated prediction. The ParCrys server predicted Npas4 to be ‘Recalcitrant to Crystallisation’. According to CRYSTALP2, Npas4 is non-crystallizable with 0.447 confidence. The probability of crystallization was also studied through FDETECT and it gave a score of 0.64 which means this protein is difficult to crystallize. PPCpred provided the crystallization probability score of 0.122 for Npas4 while the score above 0.4 means that protein has the ability to crystallize (Mizianty & Kurgan, 2011), suggesting similar difficulty in crystallization.

Protein binding regions in Npas4

Many disordered proteins bind to some other proteins and transform from disorder to order and thus perform their function (Dyson & Wright, 2016). ANCHOR predicts 10 binding sites in the disorder region of Npas4. Out of these 10 regions three of them (position 585–599, 662–720, 746–792) were present in transactivation domain of Npas4 (Table 2).

Table 2 Protein binding regions in Npas4.

Predicted disordered binding regions	
	From	To	Length	
1	362	367	6	
2	397	404	8	
3	416	422	7	
4	432	444	13	
5	464	521	58	
6	535	543	9	
7	560	573	14	
8	585	599	15	
9	662	720	59	
10	746	792	47	

Post translation modification sites

Attempts were made to predict the protein modification sites that could be occurring in Npas4. The results indicated absence of acetylation and mannosylation sites. However, two potential N-glycosylation sites were found in disorder region of Npas4 at 556 (NPTK) and 671 (NLSL) amino acid positions (Fig. 7).

Figure 7 N-linked glycosylation sites of Npas4 predicted through NetNGlyc.

X axis represents amino acid sequence from N- to C- terminal while Y axis represents scores. 0.5 is the threshold, the amino acid positions above threshold level have potential of glycosylation.

A total of 80 O’ linked glycosylation sites were present in Npas4. However, only 56 sites were found available for glycosylation according to the GlyCamserver.

According to Netphos 3.1 server predictions, 34 threonine phosphorylation sites, 53 serine phosphorylation sites and four tyrosine specific phosphorylation sites were present in Npas4 (Fig. 8). NetsurfP prediction revealed 43 serine, 28 threonine and three tyrosine residues were exposed for phosphorylation. Table S2 showed the 24 Serine, 13 Threonine and one Tyrosine kinase specific sites in Npas4. Two phosphorylation sites present (S38, S44) in bHLH domain and three sites (S98, S100 and T136) were observed in PASA domain while one site (S273) was seen in PAS B domain. Six Ser and four Thr phosphorylation sites were present in disordered transactivation domain (Table 3).

Figure 8 Potential phosphorylation sites of Npas4 predicted through NetPhos.

X axis represents amino acid sequence from N- to C- terminal while Y axis represents scores. 0.5 is the threshold, the amino acid positions above threshold level have potential of phosphorylation.

Table 3 Phosphorylation sites in disorder region of Npas4.

S.No	Phosphorylation site	Kinases	
1	Ser 611	P38MAPK, CDK5	
2	Thr 620	PKC	
3	Thr 640	CDC2	
4	Ser 656	CKII	
5	Thr 673	CKI	
6	Ser 719	P38MAPK, CKII	
7	Ser 728	CKII	
8	Ser 759	CKII	
9	Ser 777	CKII	
10	Thr 785	PKC, ATM	

Different physiological parameters of Npas4 were predicted trough Protparam server (Table 4).

Table 4 Physiological parameters of Npas4.

Molecular weight	87,116.58	
Theoretical PI	4.53	
Formula	C3894H5965N1005O1220S23	
Total no. of atoms	12,107	
Estimated half life	30 h	

Secondary structure of Npas4

The secondary structure of Npas4 showed that the protein consists of alpha helices, coils and beta sheet (Figure S3).

Tertiary structure of Npas4

The three dimensional structure of Npas4 protein has not been determined to date. Due to the absence of suitable structural template, holmology modeling can not be used (Figure S1). In order to get the high quality structure of Npas4, ab-initio and threading approaches were used. By using these approaches five models were generated through I-Tasser, 10 models through LOMET, one model from Raptor X, 10 models from MUSTER, one model from Phyre and 10 models from SPARSK-X (Table 5). All the 37 models were then subjected to validation through PROSA, Qmean and Ramachandran plot. The models with 96% residues in favoured region were further selected for refinement (Table 6, Fig. 9). All models suggested that Npas4 is a disordered protein with ordered bHLH and PAS domain. Based on Qmean, PROSA Z-score and ramachandran scores, model 9 generated LOMET was the best predicted three dimensional structure of Npas4. The possible phosphorylation sites in PASA and PASB domain of Npas4 is shown in Fig. 10.

Table 5 Models of Npas4 protein with different methods and their evaluation results.

Model/tool	Method	PROSA	Qmean	Ramachandran plot	
				Outlier	Allowed region	Favored region	
I-TASSER-1	Multiple-threading alignments	−3.6	−11.55 9.4%	18.6%	72.0%	
I-TASSER-2	Multiple-threading alignments	−1.28	−12.31 8.4%	17.4%	74.4%	
I-TASSER-3	Multiple-threading alignments	−2.59	−16.61 18.5%	28.0%	53.5%	
I-TASSER-4	Multiple-threading alignments	−4.99	−16.83 19.2%	24.6%	56.1%	
I-TASSER-5	Multiple-threading alignments	−5.07	−13.55 14.2%	27.8%	58.0%	
LEMOT-1	Local meta-threading	−2.89	−3.34 1.6%	2.2%	96.1%	
LEMOT-2	Local meta-threading	−3.07	−3.22 1.4%	2.9%	95.8%	
LEMOT-3	Local meta-threading	−3.66	−3.08	0.8%	2.4%	96.9%	
LEMOT-4	Local meta-threading	−3.72	−3.66 1.6%	2.4%	96.0%	
LEMOT-5	Local meta-threading	15.32	−10.17 10.2%	14.2%	75.6%	
LEMOT-6	Local meta-threading	−3.11	−3.96	2.0%	5.9%	92.1%	
LEMOT-7	Local meta-threading	−2.61	−2.93 1.6%	4.5%	93.9%	
LEMOT-8	Local meta-threading	−3.33	−2.75 1.5%	2.9%	95.6%	
LEMOT-9	Local meta-threading	−3.63	−2.61 0.6%	3.2%	96.1%	
LEMOT-10	Local meta-threading	3.09	−20.65 7.5%	15.9%	76.7%	
RaptorX	Threading	−5.42	−5	2.8%	4.7%	92.5%	
MUSTER-1	Multi-source threading	−1.57	−12.86	4.5%	8.2%	87.2%	
MUSTER-2	Multi-source threading	−1.51	−20.75	6.2%	12.2%	81.5%	
MUSTER-3	Multi-source threading	1.09	−2.38	7.2%	11.2%	81.2%	
MUSTER-4	Multi-source threading	−3.19	−3.91	1.8%	1.9%	96.4%	
MUSTER-5	Multi-source threading	−3.35	−2.88	1.1%	2.6%	96.2%	
MUSTER-6	Multi-source threading	−2.87	−20.70	4.8%	10.0%	85.2%	
MUSTER-7	Multi-source threading	−0.34	−14.63	4.8%	9.5%	85.8%	
MUSTER-8	Multi-source threading	−2.44	−4.14	2.1%	3.4%	94.5%	
MUSTER-9	Multi-source threading	0.53	−9.44	7.1%	12.0%	80.9%	
MUSTER-10	Multi-source threading	−1.46	−12.05	4.5%	12.9%	82.6%	
Phyre	Homology detection method	−4.13	−8.70	7.9%	12.8%	79.4%	
SPARSK-1	Fold recognition	−3.9	−9.59	5.5	7.8	86.7%	
SPARSK-2	Fold recognition	−4.57	−6.67	2.2	3.9	93.9%	
SPARSK-3	Fold recognition	−4.02	−6.12	1.5	3.6	94.9%	
SPARSK-4	Fold recognition	−5.58	−6.05	1.6	6.1	92.2%	
SPARSK-5	Fold recognition	−5.33	−7.03	1.8	4.4	93.9%	
SPARSK-6	Fold recognition	−4.77	−7.39	2.2	4.2	93.5%	
SPARSK-7	Fold recognition	−4.05	−6.53	1.4	3.6	95.0%	
SPARSK-8	Fold recognition	0.7	−11.0	6.6	7.4	86.0%	
SPARSK-9	Fold recognition	−4.3	−7.51	2.4	3.6	94.0%	
SPARSK-10	Fold recognition	−4.71	−6.99	1.8	5.0	93.2%	

Figure 9 Three dimensional structure of Npas4 predicted through MUSTER and LOMET.

(A) MUSTER-4, (B) MUSTER-5, (C) LOMET-1, (D) LOMET-3, (E) LOMET-4, and (F) LOMET-9. All predicted models show that Npas4 is a mixture of order and disorder regions.

Table 6 The validated parameters for refined models of Npas4 Protein.

Model/tool	Method	PROSA	Qmean	Ramachandran plot	
				Outlier	Allowed region	Favored region	
MUSTER-4	Multi-source threading	−3.19	−3.91	1.8%	1.9%	96.4%	
MUSTER-5	Multi-source threading	−3.35	−2.88	1.1%	2.6%	96.2%	
LEMOT-1	Local meta-threading	−2.89	−3.34	1.6%	2.2%	96.1%	
LEMOT-3	Local meta-threading	−3.66	−3.08	0.8%	2.4%	96.9%	
LEMOT-4	Local meta-threading	−3.72	−3.66	1.6%	2.4%	96.0%	
LEMOT-9	Local meta-threading	−3.63	−2.61	0.6%	3.2%	96.1%	

Figure 10 Post-translational modifications in bHLH and PAS domains.

bHLH domain is shown in blue (position 10–52 a.a), PAS A domain is shown in yellow (position 72–135 a.a) and PAS B domain is shown in red (position 216–273 a.a). Residues responsible for phosphorylation are highlighted in green.

Protein-protein interaction analysis of Npas4

To study interaction of Npas4 with other proteins, STRING database v10 (Szklarczyk et al., 2015) was used (Fig. 11). The results suggested Npas4 interaction with ARNT (Aryl hydrocarbon receptor nuclear translocator); NXF2B (nuclear RNA export factor 2B); NXF2 (nuclear export factor 2); NXF3 (nuclear export factor 3); NDNL2 (necdin-like 2); MAGEF1 (melanoma antigen family F, 1); C3orf58 (chromosome 3 open reading frame 58); NXF1 (nuclear RNA export factor 1); NUP107 (nucleoporin 107KDa); CSAG1 (Chondrosarcoma associated gene 1); MAGEF1 (Melanoma antigen family F, 1); SLC9A6 (Solute carrier family 9, subfamily A (NHE6, cation proton antiporter 6), member 6); MAGED2 (Melanoma antigen family D, 2); SLC9A9 (Solute carrier family 9, subfamily A (NHE9, cation proton antiporter 9), member 9); MAGEL2 (MAGE-like 2); SSX2B (Synovial sarcoma, X breakpoint 2B) SLC9A1 (Solute carrier family 9, subfamily A (NHE1, cation proton antiporter 1), member 1); CREB1 (cAMP responsive element binding protein 1); MAGEA1 (Melanoma antigen family A, 1) & EPAS1 (Endothelial PAS domain-containing protein 1 (also known as hypoxia-inducible factor-2alpha (HIF-2alpha).

Figure 11 Interactions of Npas4 with other proteins.

ARNT (Aryl hydrocarbon receptor nuclear translocator); NXF2B (nuclear RNA export factor 2B); NXF2 (nuclear export factor 2); NXF3 (nuclear export factor 3); NDNL2 (necdin-like 2); MAGEF1 (melanoma antigen family F, 1); C3orf58 (chromosome 3 open reading frame 58); NXF1 (nuclear RNA export factor 1); NUP88 (nucleoporin 88KDa); CSAG1 (Chondrosarcoma associated gene 1); MAGEF1 (Melanoma antigen family F, 1); SLC9A6 (Solute carrier family 9, subfamily A (NHE6, cation proton antiporter 6), member 6); MAGED2 (Melanoma antigen family D, 2); SLC9A9 (Solute carrier family 9, subfamily A (NHE9, cation proton antiporter 9), member 9); MAGEL2 (MAGE-like 2); SSX2B (Synovial sarcoma, X breakpoint 2B) SLC9A1 (Solute carrier family 9, subfamily A (NHE1, cation proton antiporter 1), member 1); CREB1 (cAMP responsive element binding protein 1); MAGEA1 (Melanoma antigen family A, 1) & EPAS1 (Endothelial PAS domain-containing protein 1 (also known as hypoxia-inducible factor-2alpha (HIF-2alpha)).

Discussion

The Npas4 gene has been in debate for more than a decade, however its structural information is scarcely available. Npas4 expression is primarily traced to neural cells, where it has been reported to be an important contributor of dendritic growth in phases of neuronal development and for modulating limbic patterning and function (Moser et al., 2004). Recently, it has been reported to be also expressed in pancreatic cells (Speckmann et al., 2016). Functionally, Npas4 is an immediate early gene, which under activation, triggers the activation of battery of genes involved in regulating brain plasticity and cognition, attributed mainly to its interaction with number of transcription factors. Npas4 belongs to the bHLH-PAS family of transcription factors and the other Npas members (1 and 3) are also linked to numerous psychiatric disorders namely autism, bipolar disorders, schizophrenia and depressive disorders (Adachi et al., 2014; Kamnasaran et al., 2003). A simulated dimerised structure of Npas4 with ARNT was reported before to discuss the potential implication of gene variants (Bersten et al., 2014). Current findings address physicochemical properties on human Npas4 protein, with a plausible 3D model, providing useful information about amino acid characteristics and possible identification of interactable proteins.

Human Npas4 is located on human chromosome 11 reference genomic contig NC000011.10, mapping to the chromosomal position 11q13.2. It possesses 11 exons that encode 2406 bp mRNA which translates into 802 amino acid long protein with 87.1 KDa molecular weight (https://www.ncbi.nlm.nih.gov/gene/266743).

In the current study the functional characterization of Npas4 has been done using in silico approaches. The MSA and phylogentic analysis showed that Npas4 is evolutionarily conserved, reflecting the highest homology to primates such as chimpanzees and monkeys.

Npas4 does not have any signal peptide hence it may not be classically secreted extracellularly. The NN score of SecretomeP also verifes Npas4 to be a non-classically secreted protein.

The physicochemical and fuctional properties of proteins are affected by post-translational modifications. Phosphorylation, acetylation and glycosylation are the most common type of post-translational modifications of proteins (Koh et al., 2012).

Acetylation is one of the major post-translational modifications (PTMs) and is important in determining the cellular localization of proteins (Qin, Pang & Zhou, 2011). There is no acetylation site present in Npas4 protein.

Glycosylation, the addition of glycosyl moiety to protein, is the most common post-translational modification in eukaryotes. N-linked and O-linked glycosylations are common while C-linked (mannosylation) is rare (Koh et al., 2012). No C-linked glycosylation sites were present in Npas4. In Npas4, two N-linked glycosylation sites were present but these sites were not available for glycosylation due to the absence of signal peptide.

Phosphorylation is another important feature contributing in PTM. Phosphorylation changes the conformation of protein making it active, inactive or modifying its function (Raghava et al., 2014). In Npas4 there were 90 potential phosphorylation sites which include 34 Threonine, 53 Serine and four Tyrosine phosphorylation sites. But not all the sites were available for phosphorylation. There were 24 Serine, 13 Theronine and one Tyrosine site available for phosphorylation.The important phosphorylation sites present in PAS A and PAS B domain were found at position 130, 136 and 273 respectively. Further experimental studies are needed to explore the role of these sites in Npas4 function.

Prediction analysis in the current study suggested presence of intrinsically disordered protein residues (IDRs) in Npas4 sequence at the C terminal, that can have profound effect on its functional versatility. Conventionally, proteins at physiological temperature exhibit a particular conformational ensemble based on optimal thermal accessibility of the ensemble to various molecular crowders and interacting proteins (Wright & Dyson, 1999). However, intrinsically disordered proteins or intrinsically disordered protein residues (IDRs) in a particular protein enables the protein to interconvert among various topological conformations (Dyson & Wright, 2005). These IDRs therefore, can provide Npas4 ease of flexibility to engage multiple targets. Moreover, IDRs can also aid in interactions by efficient utilization of less residues. This will enhance spontaneous disassociation or displacement in neuronal pathways. Previously it has been reported that structural homogeneity exhibited by IDRs is deficient which results from varied composition of Gly and Pro charged residues (Wright & Dyson, 2015), as is the case of Npas4, which may subsequently result in decreased protein folding. This can further be implicated to enhance conformational plasticity of Npas4 which may help in behaving differently to various transcription factors and enhancing its functional repertoire. The presence of IDRs C-terminal regulatory domain have also been reported in other Bhlh proteins (Fribourgh & Partch, 2017).

Moreover, previous reported behavior of IDRs suggest that the presence of more charged residues enhances electrostatic interactions which augments the propensity of post-translational modifications (Hofmann et al., 2012; Mao et al., 2010). This subsequently resulted in altered binding affinities and increased range of structural heterogeneity derived from disorder to order transitions altering protein compactness. The simultaneous consideration of Npas4 functioning is as an ‘immediate early gene’, along with all the predicted and implied consequences of harboring IDRs in its sequence. The PAS domains carrying multi-ligand binding properties can aid Npas4 to impart its role as a ‘hub protein’ in forming various complexes and key mediator in neuronal signaling.

The presence of IDR within Npas4 sequence can pose difficulty in sculpting protein structure through X-ray crystallography which may be attributable to hindrances caused by the crystal packing forces from promiscuous conformations of disordered regions (Wells et al., 2008). This could be one of the reasons that determination of Npas4 crystallization propensity was reported to be on very low scores from all servers used.

The function of protein can be best predicted if we have insight about the protein tertiary structure. The Npas4 sequence has less than 25% homology with the crystal structure of PAS domains of Npas3 and Npas1. In the absence of any structural template, ab-initio modeling approach for Npas4 structure prediction was opted. The modeled structure was composed of alpha helices and B sheets in the bHLH domain and PAS A and B domains, while the transactivation domain does not passess any tertiary structure due to the presence of IDRs. Due to the presence of proper tertiary structure of PAS domain it was involved in heterodimarization with its partner (ARNT). Current study of amino acid properties suggested preponderance of hydrophobic amino acids in both PAS A and B domains. These findings are in accordance with other bHLH proteins like Npas1, Npas3, BMAL1 and CLOCK where PAS domains are harboring internal hydrophobic cavities (Wu et al., 2016).

The Npas4 was also subjected to prediction of interacting proteins and it reflected its interaction with ARNT, which is its main partner for dimerization. The STRING analysis also showed some important interactions. The analysis showed that Npas4 functional interaction can be divided into four clusters or groups. Cluster 1 involves proteins which are responsible for forming a large protein complex (the SMC5-SMC6 complex). This protein complex is responsible for inducing double stranded DNA breaks (DSB) under neuronal stimulation, which is crucial for the expression downstream promoters such as BDNF (Madabhushi et al., 2015). Through its interaction with CDK5, which has a key role in BDNF induced dendrite development, it implicates the crucial role of Npas4 in memory plasticity through regulating the balance of inhibition in neuronal synapses. This interaction may also be related to Rett Syndrome, a disorder in which activity dependent BDNF transcription is hampered (Liang et al., 2015). Cluster 2 (Fig. 8) suggested Npas4 interact with Nuclear RNA export factor (NXF) proteins, which are RNA binding proteins, with various paralogs reported to be found in dendritic granules (Mamon et al., 2017), suggesting Npas4’s active involvement in the nuclear export of different mRNA and translational control over different proteins. Cluster 3 proteins reported to have altered expression in autism spectrum disorders (ASD), that may consequently result from knockdown of activity regulated gene transcription mediated by Npas4 (Morrow et al., 2008). Cluster 4 represented Npas4’s interaction with cyclic AMP responsive element binding (CREB) proteins. These proteins under phosphorylated state can alter chromatin structure by histone acetylation which speeds up the RNA polymerase II recruitment, triggering the gene transcriptional programs involved in synapse development (Cohen & Greenberg, 2008; Greer & Greenberg, 2008). The CREB along with Npas4 is also involved in regulating inhibitory synapses of excitatory neurons by activating BDNF transcription (Hong, McCord & Greenberg, 2008; Lin et al., 2008). Interstingly, we found three binding sites in the transactivation domain which fall in the region of IDR. This information provides a plausible rationale that CREB may be interacting with any of these residues from the transactvation domain through interaction with transcriptional coactivator, CBP which is known to cause such interaction through IDRs (Dyson & Wright, 2016).

Conclusion

Npas4 is an intrinsically disordered protein with ordered bHLH and PAS domain. The amino acid sequence analysis showed that Npas4 has a high proportion of flexible and hydrophobic amino acids that promote the disorder properties of proteins. The model 9 predicted through LOMET is the best structure of Npas4 and can be used for further analysis. This protein is difficult to crystalize, so in order to determine its tertiary structure we can use NMR and other related techniques. It has strong interactions with NXF 2B and NXF 3 proteins which implicate its potential role in cytoplasmic export of proteins from nucleus, thus influencing protein translation. This Current study also elucidates Npas4 activation with CREB proteins, suggesting Npas4’s reliance on Ca2+ dependent kinases (CDKs). The information compiled in this research can serve as useful information for identifying new drug targets, which can modulate synaptic hemostasis in neuropshychiatric and neuro-developmental disorders. Moreover, possible interaction with Nuclear RNA export factor (NXF) protein family role identification needs further elaboration. The results reported by our study are more structural and theoretical in nature, however, they may help in biophysical studies, NMR and crystallographic studies directed towards Npas4. Moreover, the considerations of results may help in future studies designed to understand binding interactions of Npas4.

Supplemental Information

Figure S1 Multiple sequence alignment of Npas4 sequence with 5SY7 (Npas3 crystal structure) and 5SY5 (Npas1 crystal structure)

The homology is shown in blue color.

Click here for additional data file.

Figure S2 Multiple sequence alignment of human Npas4 with other orthologs

Click here for additional data file.

Figure S3 Secondary structure of Npas4 predicted through PsiPred

Click here for additional data file.

Table S1 The names, Accession No and taxonomic classification of 24 different organisms whose protein sequences are used for multiple sequence alignment and phylogenetic analysis

Click here for additional data file.

Table S2 Kinase specific phosphorylation sites present in Npas4

Click here for additional data file.

Abbreviations

ARNT Aryl hydrocarbon receptor nuclear translocator

NXF2B nuclear RNA export factor 2B

NXF2 nuclear export factor 2

NXF3 nuclear export factor 3

NDNL2 necdin-like 2

MAGEF1 melanoma antigen family F, 1

C3orf58 chromosome 3 open reading frame 58

NXF1 nuclear RNA export factor 1

NUP107 nucleoporin 107 KDa

CSAG1 Chondrosarcoma associated gene 1

MAGEF1 Melanoma antigen family F, 1

SLC9A6 Solute carrier family 9, subfamily A NHE6, cation proton antiporter 6), member 6

MAGED2 Melanoma antigen family D, 2

SLC9A9 Solute carrier family 9, subfamily A NHE9, cation proton antiporter 9), member 9

MAGEL2 MAGE-like 2

SSX2B Synovial sarcoma, X breakpoint 2B

SLC9A1 Solute carrier family 9, subfamily A NHE1, cation proton antiporter 1), member 1

CREB1 cAMP responsive element binding protein 1

MAGEA1 Melanoma antigen family A, 1

EPAS1 Endothelial PAS domain-containing protein 1 (also known as hypoxia-inducible factor-2alpha (HIF-2alpha)

Additional Information and Declarations

Competing Interests

Author Contributions

Data Availability

The authors declare there are no competing interests.

Ammad Fahim and Zaira Rehman conceived and designed the experiments, performed the experiments, analyzed the data, prepared figures and/or tables, authored or reviewed drafts of the paper, approved the final draft.

Muhammad Faraz Bhatti conceived and designed the experiments, analyzed the data, authored or reviewed drafts of the paper, approved the final draft.

Amjad Ali performed the experiments, analyzed the data, contributed reagents/materials/analysis tools, authored or reviewed drafts of the paper, approved the final draft.

Nasar Virk and Amir Rashid conceived and designed the experiments, contributed reagents/materials/analysis tools, prepared figures and/or tables, authored or reviewed drafts of the paper, approved the final draft.

Rehan Zafar Paracha analyzed the data, contributed reagents/materials/analysis tools, authored or reviewed drafts of the paper, approved the final draft.

The following information was supplied regarding data availability:

The raw data are provided in the Supplemental Files.

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
