# Peer review of "Structural insights and characterization of human Npas4 protein"

_PeerJ, doi:10.7717/peerj.4978_

## Round 0.1 · original submission · Major Revisions

Although one of the reviewers recommended rejection, I decided to give you an opportunity to answer critiques and revise manuscript. Please make sure that you addressed all critical points raised by the reviewers and adjust the manuscript accordingly.

Reviewer 1 ·

Basic reporting

This report focuses on the activity-dependent transcription factor NPAS4. The authors attempt to provide structural insight and to predict proteomics information regarding this protein. The goals of this study are clearly stated. However, this manuscript does not provide sufficient information for this topic. This reviewer lists the following 4 concerns:

1. There are grammar errors throughout the manuscript. There are also terms that need clarification. For instance, in line 50-51, the "activity dependent genes" is confusing.
2. The authors did not provide sufficient background information. For example, the authors mentioned that there is no structural homologues for homology modeling of NPAS4. However, in Protein Data Bank (PDB), at least a crystal structure of the heterodimeric NPAS3-ARNT complex with HRE DNA is available. Or the authors should have describe that why this crystal structure is irrelevant for NPAS4 modeling.
3. Except the figure 9, none of these figures contain proper labels to be informative. The structure model is not further analyzed to provide insights.
4. The set of results is meaningful. However, the data is too preliminary. The results generated in this report are not sufficient to facilitate future studies.

Experimental design

The authors clearly stated the absence of structural information for NPAS4. However, solely
relying on web servers, the authors have not developed any methods or designed any experiments to generate results that contain unique insights. Moreover, as mentioned above, the authors did not mention why homology modeling using the crystal structure of NPAS3 is infeasible. It is difficult to put this manuscript into the category of Research Articles or Literature Review Articles, the only two types of articles PeerJ considers.

Validity of the findings

Since all the results reported in this manuscript were generated by web servers, they are too preliminary and thus difficult to validate.

Reviewer 2 ·

Basic reporting

In their manuscript submitted to PeerJ, Fahim et al present an in-silico study of an important human neuro-transmission Npas4 protein. They have reported a tentative tertiary structure and analyzed physio-chemical properties of Npas4 protein along with study of its closest phylogeny and identified the interaction partners of the protein.
Introduction has been well-written with adequate references emphasizing on the importance of Npas4 and the objective of this research work. However, the paper needs rephrasing at several places as mentioned below to make the comprehension more easily understandable:
• L61: “predict the structural insight” should be replaced with “provide structural insights into”
• L69, L73: “orthologus” spelling is incorrect
• L77: “forecast” should be replaced with “calculate” or “predict”
• L94, L97: “Accession No.” is misleading
• L100: “,” should be replaced with “.”
• L106: Line needs rephrasing
• L141: “.” should be removed after “potential”
• L144: “are” to inserted before “present”
• L148: correct spelling as “revealed”
• L157: “,” to be replaced by “and”
• L160: “are” to be inserted before “shown”
• L187: Replace “,” with “.”
• L189, L194, L195, L196: “Npas4” should be replaced with “Npas4” as only gene names are written in italics.
• L199: replace “affect” by “are affected by”
• L213: “phosphorylation,” to be replaced by “phosphorylation.”
• L221: Line needs rephrasing
• L252: the word “elaboration” is confusing
The manuscript has been drafted in accordance with PeerJ standards and all the figures with the exception of Fig 3 are of high quality. Some additional suggestions that might benefit a reader are: 1) adding a color legend in Fig 4 that explains the red bar and purple straight lines, and 2) adding the sequence length of each domain along with highlighting the N- and C- terminal regions in Fig 7.
The raw data has not been shared adequately. UniprotID corresponding to the Npas4 protein sequence should be mentioned in the first section of Materials and Methods. Also, the parameters used for running ClustalX and Mega7 are lacking in the manuscript. Tuning the parameters always changes the output results. So mentioning parameter values or mentioning default parameters in Experimental section is appreciable.

Experimental design

The research problem as posed by the authors is very relevant in the absence of sufficient literature regarding structural insights and characterization of human Npas4 or its homologues. But, their experimental design and approach to address their goal of research lacks technical expertise and is not up to the scope of this journal.
The main drawback of the manuscript is the ambiguity in the results obtained by running various bioinformatics servers used by the authors. Uniprot database provides a detailed analysis of Npas4 protein (Uniprotid: Q8IUM7) with a list of similar proteins from phylogenetic analyses clustered on basis of sequence identities, yields information on interaction partners of this protein from STRING and other databases and also provide models of this protein from various 3D structure databases. Its therefore, crucial that the authors provide a comparative explanation of how their results are different or superior than those provided by Uniprot.
As with all prediction servers, it’s also highly recommended to assess the statistical significance of the results, which the authors have failed to address in the current manuscript. For example, while discussing the results of tertiary structural prediction, authors should provide more details of I-tasser output, like the values of C-score for 5 models and templates selected by I-tasser to generate such models. It’s equally important to analyze the templates and check alignment accuracy of the template with Npas4 protein (Tm score). Also, there is no mention of how clusters were generated from the output of STRING database to determine interaction partners.

Validity of the findings

The results as reported by authors need several edits as suggested below:
1) L105: Authors mention using Multiple Sequence alignment but there is no accompanying figure of the alignment highlighting the conserved residues.
2) L108: Supplementary Table S1 is missing.
3) L109: Use of taxonomic nomenclature is preferred instead of the generic animal names like chimpanzees and monkey.
4) L109: Please check spelling of rodents like mramrt.
5) Fig3: Resolution of the plots needs to be greatly improved.
6) L115-137: Authors have reported values of several physiochemical properties of Npas4 but it will be interesting to see how these values are correlated to the structure/functioning of the protein. For example, does the flexible amino acid residues from position 204 to 295 as identified by flexibility index correlate with flexible region in the modelled structure obtained from I-tasser.
7) L153-155: While predicting secondary structure from GOR4, PsiPred and JPred3, it will be helpful if the authors provide a consensus secondary structure figure representing which residues/regions of Npas4 correspond to helix, sheet and coil.
8) L156-160: Authors have chosen the highest C-score model from I-Tasser to be the best model. However, they haven’t provided any C-score values nor provided the statistics of ProSA, PROCHECK and Pymol to compare these 5 models. It will be also good to compare the templates used by I-tasser and those used by 3D structure databases as in Uniprot to have better confidence about the tertiary structure of Npas4 protein model.
In the discussion section of the manuscript, the physicochemical property values and study of posttranslational modification sites is interesting but mostly based on prediction without any correlation to the tertiary structure. The significance of different interacting protein clusters of Npas4 as obtained from STRING database is well documented but authors are encouraged to discuss the input parameters used (like maximum number of interactors to be shown in 1st and 2nd shell) to explain the analysis. A few corrections needed are:
1) L185: Correct the reference author name as Adachi et al 2014.
2) L187: The reference (Katoh et al) doesn’t match with the paper cited.
3) L222: STRING is a database and it’s advisable to write in capitals so as to not to confuse readers with word “string”.

Additional comments

Overall, the authors have well highlighted the scientific importance of Npas4 and the purpose of their study. However, their results fail to provide detailed structural insights of human Npas4 protein as claimed in their manuscript title and hence, needs further research work and statistical analyses, before Acceptance.

---

## Round 0.2 · accepted · Accept

Since reviewer #1 (who recommended rejection of the original submission) did not find any issues with the revised manuscript, it is clear that you did a good job addressing critical issues raised by the reviewers and revising the manuscript.

Reviewer 1 ·

Basic reporting

no comment

Experimental design

no comment

Validity of the findings

no comment

Additional comments

Thank you for providing me the opportunity to re-evaluate this manuscript.

The two major concerns I had for the previous manuscript were: 1. the lack of unique insight provided by the authors to facilitate future studies; 2. the authors did not address why homology modeling is infeasible.

I recommend acceptance of the revised manuscript for two reasons: 1. the authors actively provided reasonable response to my concerns. 2. After more investigation of this topic, I better understand that Npas4 protein is not only important, but also very unknown at structural level. Even the results author provided does not suffice to lead clear direction, they encourage more studies in this topic.